Geographic patterns of distribution and ecological niche of the snake-necked turtle genus Hydromedusa

Muller Márcia M.P. 1
Santana Diego J. jose.santana@ufms.br 2
Costa Henrique C. 3
Ceron Karoline 4
1 Instituto de Biociências, Letras e Ciências Exatas, Universidade Estadual Paulista , São José do Rio Preto , São Paulo , Brazil
2 Instituto de Biociências, Universidade Federal de Mato Grosso do Sul , Campo Grande , Mato Grosso do Sul , Brazil
3 Departamento de Zoologia, Universidade Federal de Juiz de Fora , Juiz de Fora , Minas Gerais , Brazil
4 Departamento de Biologia Animal, Universidade Estadual de Campinas , Campinas , São Paulo , Brazil
Sunny Armando
Electronic publication date: 2024 Mar 26
Publication date: 2024
Volume: 12
Electronic Location ID: e16712
Received 2023 May 15; Accepted 2023 Dec 1
Copyright: ©2024 Muller et al.
Copyright year: 2024
Copyright holder: Muller et al.
License: This is an open access article distributed under the terms of the Creative Commons Attribution License, which permits unrestricted use, distribution, reproduction and adaptation in any medium and for any purpose provided that it is properly attributed. For attribution, the original author(s), title, publication source (PeerJ) and either DOI or URL of the article must be cited.
License URL: https://creativecommons.org/licenses/by/4.0/

Keywords: Overlap, Species distribution modeling, Temperature

Funding: Coordenação de Aperfeiçoamento de Pessoal de Nível Superior—Brasil (CAPES)—Finance Code 001 Coordenação de Aperfeiçoamento de Pessoal de Nível Superior and Conselho Nacional de Desenvolvimento Científico e Tecnológico CAPES, Finance Code 001 CNPq 402012/2022-4 CNPq CNPq 309420/2020-2 CNPq 402012/2022-4 São Paulo Research Foundation-FAPESP 2020/12588-0 This work was supported by the Coordenação de Aperfeiçoamento de Pessoal de Nível Superior—Brasil (CAPES)—Finance Code 001. Márcia Muller got her scholarship from the Coordenação de Aperfeiçoamento de Pessoal de Nível Superior and Conselho Nacional de Desenvolvimento Científico e Tecnológico (CAPES, Finance Code 001; CNPq 402012/2022-4). Diego J. Santana has research fellowships from CNPq (CNPq 309420/2020-2; CNPq 402012/2022-4). Karoline Ceron is funded by São Paulo Research Foundation-FAPESP (Grant 2020/12588-0). The funders had no role in study design, data collection and analysis, decision to publish, or preparation of the manuscript.

==============================
Biotic and abiotic factors play a crucial role in determining the distribution of species. These factors dictate the conditions that must be met for a species to thrive in a particular area. Sister species that present some degree of niche overlap can shed light on how they are distributed and coexist in their environment. This study aims to investigate the geographical distribution and ecological niche of the sister species of snake-necked turtles Hydromedusa maximiliani and H. tectifera. By analyzing their niche overlap, we aim to obtain a better understanding of how these two species coexist and which variables are determining their occurences. We applied species distribution modeling and compared the niches using the niche equivalence and similarity tests. Our findings show that the distribution of H. maximiliani is most influenced by temperature seasonality and isothermality, while H. tectifera is most affected by the temperature seasonality, precipitation of warmest quarter and mean diurnal range. In addition, our results suggest that the niche expressed by H. maximiliani retained ecological characteristics that can accurately predict the H. tectifera distribution, but the inverse is not true. In this sense, differences are not solely due to the geographic availability of environmental conditions but can reflect niche restrictions, such as competition.

Introduction

The distribution patterns of species can be shaped by abiotic factors such as climate and elevation, which constitute the Grinnellian niche of species (Garzón, Dios & Ollero, 2008; McVicar & Körner, 2013), as well as biotic factors, including interactions between the same trophic level such as competition, and between different trophic levels such as host-parasite/host-pathogen relationships, which constitute the Eltonian niche of species (Soberón, 2007; Wisz et al., 2013; González-Salazar, Stephens & Marquet, 2013; Acevedo et al., 2014). The presence of closely related species in the same area can also influence distribution patterns, for example, habitat partitioning when species differ in traits such as substrate selection, trophic niche, or period of activity (Nogueira et al., 2019; Ceron et al., 2021). These factors collectively determine the species’ ecological niche, i.e., the specific environmental conditions required for the establishment of a species in a particular region (Lewis et al., 2017; Sales, Hayward & Loyola, 2021).

Species distribution modeling (SDM) can provide insights into the important abiotic factors influencing species distribution (i.e., Grinnellian niche) and even predict potential occurrences in unsampled areas or areas that may be at risk of invasion (Guisan & Thuiller, 2005). These characteristics turned the SDM into an essential tool for mapping and predicting distribution shortfalls for a given species or a set of species, and verifying the similarity between niches of different species (Guisan & Thuiller, 2005; Guisan, Thuiller & Zimmermann, 2017). The investigation of niche overlap among populations or taxa that occur in the same place at the same time (sympatry) is a fundamental aspect of ecological research, as it can shed light on how species coexist and occupy their environment in a given area (Marko, 2008; Cavalcante et al., 2020). In addition to sympatry, the shared evolutionary history can increase competition and make coexistence difficult. As related species tend to have similar ecological preferences, showing niche conservatism, it has long been recognized that the phylogenetic relationships between species may influence their coexistence, such as sister species (Wiens et al., 2010). Despite the potential niche overlap, sister species may exhibit differences in the specific ecological characteristics they use to occupy similar niches (Duré, Kehr & Schaefer, 2009). However, failure to exhibit such differences can lead to competitive exclusion or character displacement (Brown & Wilson, 1956; Hardin, 1960). In this sense, resource partitioning in one way or another is a requirement, if competing species are to coexist sympatrically under resource-limited conditions (Mac Arthur, 1972). Therefore, studying niche overlap can provide valuable information on the factors driving sister species’ distribution and coexistence.

The South American freshwater snake-necked turtle genus Hydromedusa is composed of two species with partially sympatric geographic ranges: Hydromedusa maximiliani (Mikan, 1825) and H. tectifera Cope, 1870 (Fig. 1), both occurring in the Atlantic rainforest. Hydromedusa maximiliani primarily occurs in the eastern and southeastern regions of Brazil, associated with low-depth watercourses that have clean, cool water, and sandy or stony substrates, mostly at high elevations (Souza & Martins, 2009), while H. tectifera is found in Argentina, Brazil, Paraguay, and Uruguay, in lentic and lotic environments, including clean water, urban rivers with sandy bottoms, and even polluted water (Ribas & Monteiro-Filho, 2002; Sánchez et al., 2019; Alcalde, Sánchez & Pritchard, 2021). Sympatry between these species is known in southeastern Brazil, and when it occurs H. maximiliani occupy areas above 600 m above sea level (asl), with H. tectifera occurring at lower elevations (Souza, 2005).

Figure 1 Adult individuals of (A) Hydromedusa maximiliani from Parque Estadual Carlos Botelho, São Paulo, Brazil, and (B) H. tectifera from Bom Jardim de Minas, Minas Gerais, Brazil.

Photo credit: Karoline Ceron (A) and Diego José Santana (B).

Although both Hydromedusa species currently show partially sympatric ranges, their speciation history is unknown, meaning that their sympatry is not evidence of sympatric speciation (when an ancestral population splits without any geographic isolation (Futuyma & Kirkpatrick, 2017)). The Atlantic rainforest has a complex and rich biogeographic history due to river basins, mountain chains, and past climatic fluctuations responsible for several cases of allopatric speciation (populations of an ancestral species geographically separated by a physical barrier (Futuyma & Kirkpatrick, 2017)) (e.g., Pavan, Narvaes & Rodrigues, 2001; Morales et al., 2020; Barbo, Nogueira & Sawaya, 2021) and the current geographic overlap between H. maximiliani and H. tectifera may be result of range expansions after speciation.

Given the potential niche overlap between these sister species, this study aims to investigate the geographical distribution and ecological niche of the snake-necked turtles H. maximiliani and H. tectifera. In this sense, we would expect a different response to environmental variables by each species allowing a different geographic range with some degree of overlap, as stated by the niche conservatism hypothesis (Wiens et al., 2010). Thus, by analyzing niche overlap and performing niche tests, we intend to have a better understanding of how these two species coexist and which variables determine their occurrences.

Material & Methods

Data source

To gather information on the distribution of H. maximiliani we used a dataset from the “Plano de Ação Nacional para a Conservação da Herpetofauna Ameaçada da Serra do Espinhaço em Minas Gerais” (https://www.gov.br/icmbio/pt-br/assuntos/biodiversidade/pan/pan-herpetofauna-do-espinhaco), which was compiled by one of us (H.C.C.) through a combination of literature records (e.g., Costa et al., 2015) and expert-verified records obtained from citizen science (iNaturalist). For H. tectifera, information on distribution was compiled from relevant literature sources (e.g., Sánchez et al., 2019). When coordinates were not available from the primary sources, we conducted searches based on locality names. If the locality name was ambiguous or could not be found, we excluded the point from our database. We also included data from a specimen of H. tectifera captured by us during a fieldwork in the municipality of Bom Jardim de Minas, state of Minas Gerais, Brazil in November 2021 (collection permit SISBIO #72874-3 issued by the Instituto Chico Mendes de Conservação da Biodiversidade).

Species distribution modeling (SDM)

Two sets of environmental predictors, namely climate and slope, were used in this study. We downloaded 19 bioclimatic variables from the CHELSA Climate version 2.1 database at a resolution of 30 s (Karger et al., 2017), averaged over the 1981–2010 period. The slope variable was based on the digital elevation models from global 250 m GMTED2010 and near-global 90 m SRTM4.1 dev, at a resolution of 3-arc seconds (Amatulli et al., 2018). The slope was resampled to a resolution of 30 arc-sec using the nearest neighbor interpolation in the ‘raster’ package (Hijmans & Van Etten, 2016).

Despite the long-standing and important role of SDMs in ecological research, we acknowledge that correlative approaches present important shortcomings that challenge their applicability in a highly dynamic world. These shortcomings are mainly related to the data used and the applied methodology, which may result in biased results (see Anderson, 2012 for a review). To deal with it, we employed several procedures to minimize biases in data and modeling approach. Thus, the procedures employed to minimize overprediction and multicollinearity followed Ceron et al. (2023). Specifically, to minimize overprediction and low specificity, we cropped the environmental layers to span from latitude −90 to −30 and longitude −50 to 15 (values in decimal degrees). To address autocorrelation among occurrence data and the potential for overfitting issues, we used the package ‘spThin’ (Aiello-Lammens et al., 2015) to eliminate one of each pair of records falling within single grid cells (∼5 km). To mitigate multicollinearity among the environmental explanatory variables, we calculated the Variance Inflation Factor (VIF) values for each species. Variables with high correlation (VIF > 5) were removed through a stepwise procedure, using ‘usdm’ package (Naimi, 2013). As a result, we retained six and seven variables for H. maximiliani and H. tectifera models, respectively.

In addition, species distribution modeling was performed as previously described in Ceron et al. (2023). In this sense, we employed nine different algorithms implemented in the ‘biomod2’ package (Thuiller et al., 2016) in R environment (R Core Team, 2022). These included three regression methods (GAM: general additive model (Hastie & Tibshirani, 1990); GLM: general linear model (McCullagh & Nelder, 1989); and MARS: multivariate adaptive regression splines (Friedman, 1991)), three machine learning methods (GBM: generalized boosting model (Ridgeway, 1999) MAXENT: Maximum Entropy (Phillips, Anderson & Schapire, 2006); and RF: random forest (Breiman, 2001)), two classification methods (CTA: classification tree analysis (Breiman, 1984); and FDA: flexible discriminant analysis (Hastie, Tibshirani & Buja, 1994)), and one envelope model (SRE: Surface Range Envelop (Booth et al., 2014)). To ensure the absence (or pseudo-absence) data for most of these models (except SRE), we generated two sets of random pseudo-absence (PA) points, each with the same size as the sets of true presences, across the model background (1,000 PA points in each set). When using regression algorithms (e.g., GLM and GAM), the method used to select pseudo-absences had the greatest impact on the model’s predictive accuracy. Usually, more accurate results within these methods were obtained when a large/moderate number of pseudo-absences per replicate were used (Barbet-Massin et al., 2012). However, Liu, Newell & White (2019) based on several simulations using dozens of algorithms, including regression methods, conclude that a large number of random points (NRP—representing pseudo-absences) is not always an appropriate strategy. In most of these situations, a few random points as pseudo-absences perform better than many random points (>5,000), especially when fewer presences are available, as in our case. The models were calibrated using 70% of randomly selected data and the other 30% of the data was used for intrinsic model evaluation.

To assess the performance of individual models, we used two metrics, namely the true skill statistic (TSS) and the area under the curve of receiver operating characteristics (ROC), as implemented in the biomod2 R package. TSS is calculated as “sensitivity + specificity − 1” and ranges from −1 to +1, with +1 indicating perfect agreement, 0 implying agreement expected by chance, and values less than 0 indicating agreement lower than expected by chance. We selected models with high predictive accuracy (TSS > 0.8) for the projection of Hydromedusa distribution. We constructed ensemble maps based on the median of two runs of all the selected models in which individual accuracy had TSS value equal to or greater than 0.8. The variability in the performance of modelling techniques, as well as the influence of the species presence data, led several researchers to recommend an ensemble modelling approach (e.g., Araújo & New, 2007; Comte & Grenouillet, 2013). To assess the importance of variables in the ensemble prediction, we employed a permutation procedure, as described by Thuiller et al. (2016).

Niche comparisons

We initially used all bioclimatic predictors to conduct niche equivalence/similarity tests, and principal component analysis (PCA-env method) (Broennimann et al., 2012). The PCA-env employs a principal component analysis (PCA) to reduce the dimensionality of environmental data to the first two main axes, using full background data as calibration. Subsequently, the PCA-env compares the full background data to the areas effectively occupied by species in their respective ranges. To generate smoothed densities of occurrences and environmental availability, we employed Kernel density functions following the approach outlined by Broennimann et al. (2012).

To quantify the degree of niche overlap between Hydromedusa species, we calculated Schoener’s D statistic directly within the ecological niche space, as outlined by Schoener (1968) and Warren, Glor & Turelli (2008). The D statistic varies from 0 to 1, indicating no overlap (0) to complete overlap (1). To evaluate whether the ecological niches of Hydromedusa are significantly distinct from one another and whether the two niche spaces are interchangeable, we conducted a niche equivalence test, following the methodology of Warren, Glor & Turelli (2010). This involved comparing the niche overlap values (D) for H. tectifera and H. maximiliani to a null distribution generated from 100 overlap values. This approach is known to provide a high level of confidence in rejecting the null hypothesis, as demonstrated by Hanley & McNeil (1982). The niche equivalence test is conservative and considers the exact locations of the species without accounting for the surrounding environmental space. We considered ecological niches to be non-equivalent if the niche overlap value of the species being compared was significantly lower than the overlap values from the null distribution (P ≤ 0.05).

Furthermore, we conducted a niche similarity test, which accounts for differences in environmental conditions in geographic areas where both species are distributed (Warren, Glor & Turelli, 2010). The concept of niche similarity tests was evaluates whether niche models calibrated for one species can accurately predict occurrences of other species beyond what would be expected by chance (Peterson, Soberón & Sánchez-Cordero, 1999). A significant difference from the niche similarity test would not only indicate differences in the environmental niche space occupied by the two species, but also that these differences are not solely due to the geographic availability of environmental conditions. These analyses were performed using R (R Core Team, 2022) with the ‘ecospat’ package (Di Cola et al., 2017).

Results

A total of 101 distribution records were obtained for H. maximiliani and 127 for H. tectifera (Table S1). The distribution of H. maximiliani comprehends eastern Brazil, encompassing the states of Bahia, Minas Gerais, Espírito Santo, Rio de Janeiro, and São Paulo, at elevations ranging from 4 to 1,499 m above sea level (Fig. 2). Hydromedusa tectifera is distributed across elevations ranging from sea level to 1,295 m and is found in four countries: Brazil (states of Minas Gerais, Rio de Janeiro, São Paulo, Paraná, Santa Catarina, and Rio Grande do Sul), Argentina (provinces of Misiones, Entre Ríos, Buenos Aires, Santiago del Estero, Córdoba, Salta, and San Luis,), Uruguay (departments of Salto, Tacuarembó, Cerro Largo, Río Negro, Soriano, Maldonado, and Montevideo), and Paraguay (departments of Alto Paraná, Guairá, and Itapúa) (Fig. 2). The geographic ranges of the two species overlap in southeastern Brazil (states of São Paulo, Minas Gerais, and Rio de Janeiro) (Fig. 2).

Figure 2 Topographic map showing the known records of the two species of the snake-necked turtle genus Hydromedusa in South America, where yellow circles, H. tectifera; blue squares, H. maximiliani.

DJ Santana prepared the map using QGIS 3.8.

The resulting ensemble models yielded averages of TSS = 0.98 and ROC = 0.99 for the H. maximiliani model. The predicted distribution of H. maximiliani remained adjusted to the occurrence points in eastern Brazil, plus a projected possible occurrence in northeastern Brazil (Fig. 3). Temperature seasonality was the most import variable (39% of explanation) for H. maximiliani distribution, followed by isothermality (29% of explanation) (Table S1). Other climate predictors contributed to the explanation of H. maximiliani ecological niche models, albeit to a lesser extent.

Figure 3 Species distribution modeling from ensemble projections for the snake-necked turtle genus Hydromedusa.

(A) H. maximiliani, (B) H. tectifera, and (C) overlap distribution of climatic niche of both species. K Ceron ran the species distribution modeling in R software, and DJ Santana prepared the map using QGIS 3.8.

The resulting ensemble models yielded averages of TSS = 0.94 and ROC = 0.99 for H. tectifera model. The predicted distribution of H. tectifera remained adjusted to the occurrence points in southeastern and southern Brazil, Uruguay, eastern Paraguay, and eastern Argentina, plus a projected possible occurrence in northeastern Brazil (Fig. 3). As H. maxilimiliani, temperature seasonality was the most import variable (61% of explanation) to H. tectifera distribution, followed by precipitation of warmest quarter (23% of explanation) and mean diurnal range (22% of explanation) (Table S1). Other climate predictors contributed to the explanation of H. tectifera ecological niche models, albeit to a lesser extent.

Niche overlap results suggest a small overlap in the environmental space inhabited by Hydromedusa species (D = 0.22), ruling out the hypothesis of niche equivalence between species (p > 0.05). However, when analyzing the niche similarity between species, the niche was found to be similar when comparing the niche of H. maximiliani to the background of H. tectifera (p = 0.01), implying that H. maximiliani niche is more similar in its environmental distributions than expected given by H. tectifera respective ranges. However, when comparing the background of H. maximiliani to H. tectifera we did not reject the null hypothesis (p = 0.35) (Fig. 4), indicating that the niches were no more similar to each other than expected by chance.

Figure 4 Niche filling of H. tectifera (green) and H. maximiliani (blue) depicted using 50% and 100% kernel density estimation (indicated by dashed lines and straight lines, respectively).

The niche filling for each species is presented in relation to the available background environment.

Discussion

We found a small overlap in the ecological niche of the two South American snake-necked turtles, Hydromedusa maximiliani and H. tectifera (Fig. 4). Hydromedusa maximiliani occurs primarily in mountainous regions (4–1,499 m asl) of eastern Brazil. In contrast, the distribution of H. tectifera spans southeastern and southern Brazil, extending southward to Uruguay, and southwestward to Argentina and Paraguay. This biogeographic scenario indicates an allopatric distribution pattern, consistent with some observations in phylogenetically related species (e.g., Costa et al., 2008; Vargas-Ramírez et al., 2020). One potential factor that could explain the different spatial occurrences of these two species is their habitat preferences. While H. maximiliani is typically found in lotic environments such as in mountain streams (Souza & Martins, 2009), H. tectifera is found in lentic environments, but can also occur in lotic environments (Sánchez et al., 2019). Additionally, H. tectifera appears to withstand human changes in the environment better than H. maximiliani (Famelli et al., 2012; Semeñiuk et al., 2019). This difference in habitat preference and potential adaptability to better survive in anthropic areas may explain why H. tectifera has a larger niche background compared to H. maximiliani, as it is able to occupy a wider range of environments.

Our species distribution modeling (SDM) analyses revealed that different environmental variables are important for explaining the distribution of H. maximiliani and H. tectifera. For H. maximiliani, temperature seasonality and isothermality were the main explanatory variables. High values of isothermality suggest that areas with minimal temperature variations between day and night compared to variations along the year (Ceron et al., 2021) may be selected by Hydromedusa maximiliani. Additionally, the temperature seasonality has been shown to be important for this species once its activity is directly related to temperature. Hydromedusa maximiliani is a thermoconformer ectotherm, inhabiting forest streams with year-round cold waters, rarely basking (Souza & Martins, 2006). The seasonality of rainfall and temperature influence seasonality in ectotherms as H. maximiliani, whose specimens are more active in warmer months (Souza & Abe, 1997; Souza, 2004; Famelli et al., 2014).

The distribution of H. tectifera is mainly influenced by temperature seasonality, precipitation of warmest quarter, and mean diurnal range. These variables are likely linked to different aspects of H. tectifera biology. Temperature seasonality and mean diurnal range (mean of monthly maximum temperature minus minimum temperature) may affect H. tectifera behavior, as it shows seasonal activity, with peaks in spring and summer (warmer months), and is a thermoconformer, suffering direct influence of water temperature in body temperature (Molina & Leynaud, 2017). In a multi-taxa study, mean diurnal range was found to have the highest overall explanative power across turtle species, reflecting the strong dependence of temperature shaping species ranges (Ihlow et al., 2012). Temperature seasonality plays a critical role in hatching success for turtles, as low temperatures can slow down embryonic development and increase predation risk (Souza & Vogt, 1994; Ji et al., 2003). Higher temperatures can increase metabolic demand (Haskins & Tuberville, 2022), leading to higher rates of growth, reproduction, and movement (Ji et al., 2003; Li et al., 2021). However, if temperatures exceed the optimal range for a species, it can result in metabolic problems, disrupting an individual’s physiology and interfering with their natural biology (Zhang et al., 2019; Li et al., 2021). This could be the case for H. tectifera, which appears to have physiological mechanisms that enable it to maintain activity levels from 9–25 °C (Molina & Leynaud, 2017). Influence of the precipitation of warmest quarter in the geographic range of H. tectifera could be explained because freshwater turtles use water as a thermal buffer (Waterson et al., 2016). Additionally, precipitation increases water availability (Rodrigues et al., 2018) and make favorable habitats for aquatic vegetation available, offering shelter and establishing a structured microhabitat with greater availability of food resources (Taigor & Rao, 2010). However, excessive rainfall can also result in flooding, destroying nests (Bager & Rosado, 2010) and displacing turtles from their usual habitats. Some authors have suggested that specimens of H. tectifera may bury themselves in the mud when the water body dries out. But additional evidence is required to support this claim (Alcalde, Sánchez & Pritchard, 2021).

Our study revealed some similarities with a previous study on niche modeling for H. maximiliani (Costa et al., 2015). The locality records of H. maximiliani in our study approach the same locations as those in the states of Bahia, Minas Gerais, Espírito Santo, Rio de Janeiro, and São Paulo, as observed in Costa et al. (2015). However, the variables that influenced the distribution of H. maximiliani differed between the two studies. For Costa et al. (2015), the variables that most influenced the species’ distribution were annual mean temperature and mean diurnal range. In contrast, our study found that temperature seasonality and isothermality were the most influential variables. The difference in findings may be attributed to the methods used in the studies, as each employed different numbers of algorithms to their respective models and/or had varying sample sizes (with a difference of 53 samples between both studies). Nevertheless, temperature was a common variable that explained the distribution in both studies, which may be due to the thermoconformity strategy employed by this species (Souza & Martins, 2006).

Regarding the distribution of H. tectifera, there are disjunct populations in the westward regions of Argentina. Although we performed species distribution modeling, the origin of these populations remains unclear. To better understand the nature of these isolated populations, further research is needed, focusing on the phylogeography of the species. This involves examining how past climatic conditions have influenced the current distribution of the species, taking into account not only climatic features but also geomorphological and genetic data (e.g., Werneck et al., 2012; Oliveira et al., 2018; Carvalho et al., 2023).

Phylogenetically related species can sometimes occupy similar climatic niches (Rodrigues et al., 2018), but in different regions or geographic areas (Kozak & Wiens, 2006), separated by barriers, such as mountain range or a river basin (Vargas-Ramírez et al., 2020; De Oliveira et al., 2021). This is true when we access the niche similarity test of H. maximiliani to the background of H. tectifera. Our results suggest that the niche expressed by H. maximiliani retained ecological characteristics (niche conservatism) that can accurately predict the geographic range of H. tectifera. When geographic ranges are contiguous and niches are similar, differentiation exists, but it is not manifested in niche characteristics (Tocchio et al., 2015). But when comparing the background of H. maximiliani to H. tectifera niche, no similarity is found between niches. In this sense, differences are not solely due to the geographic availability of environmental conditions but can reflect niche restrictions, such as competition. For example, in a study with sister species of hares (Lepus spp.) in Italy, Acevedo et al. (2014) found that island populations of L. corsicanus are evolving in the absence of potential competitors and are displaying what could closely resemble the part of its fundamental niche. In contrast, in continental populations, which have evolved in contact with competitor species, the pattern of distribution explained by the models is closer to the species’ realized niche (Acevedo et al., 2014). The same can be hypothesized for Hydromedusa species, with H. tectifera sharing its distribution range with other freshwater turtles such as Acanthochelys spixii and Phrynops hilarii (Alcalde, Sánchez & Pritchard, 2021), which can impose some niche restrictions on H. tectifera populations, not reflecting H. maximiliani niche. Morphological differences between Hydromedusa maximiliani and H. tectifera also may be related to their differences in spatial occurrence. While Hydromedusa maximiliani has a flatter carapace, which gives it better hydrodynamic ability in lotic environments (Stayton, 2011), H. tectifera has a higher carapace. This difference in morphology could result in the two species occupying different spaces within their shared environment, as they are adapted to different ecological niches (Maltseva et al., 2021).

Conclusions

Our study reveals that Hydromedusa maximiliani and H. tectifera generally exhibit an allopatric distribution pattern, with a small overlap in their distributions. H. maximiliani prefers colder mountainous regions with lotic environments, while H. tectifera occurs in a broader range of ecoregions with both lentic and lotic environments. The distribution modeling analysis has shown that different environmental variables are important in explaining the distribution of H. maximiliani and H. tectifera. Within their shared environment, the two species showed small differentiation in space use, which allow their coexistence. However, niche differences between them are not driven only by the geographic availability of environmental conditions, but also by biotic interactions that may impose niche restrictions. Future research investigating the phylogeography of Hydromedusa can aid in the understanding of how past climate conditions impacted their present-day distribution, and the establishment of isolated populations.

Supplemental Information

Table S1 Coordinates, altitude and localities from points used in the study

H. maxim iliani (Costa, H.C.), H. tectifera (Sánchez et al., 2019) and one individual of H. tectifera collected by us in a fieldwork.

Table S2 Range of climatic variables to Hydromedusa species

Additional Information and Declarations

Competing Interests

Author Contributions

Field Study Permissions

Data Availability

The authors declare there are no competing interests.

Márcia M.P. Muller conceived and designed the experiments, performed the experiments, analyzed the data, prepared figures and/or tables, authored or reviewed drafts of the article, and approved the final draft.

Diego J. Santana conceived and designed the experiments, analyzed the data, prepared figures and/or tables, authored or reviewed drafts of the article, and approved the final draft.

Henrique C. Costa conceived and designed the experiments, authored or reviewed drafts of the article, and approved the final draft.

Karoline Ceron conceived and designed the experiments, performed the experiments, analyzed the data, authored or reviewed drafts of the article, and approved the final draft.

The following information was supplied relating to field study approvals (i.e., approving body and any reference numbers):

Field collections were approved by Instituto Chico Mendes de Conservação da Biodiversidade.

The following information was supplied regarding data availability:

The raw data are available in the Supplementary File.

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
