# Peer review of "Geographic patterns of distribution and ecological niche of the snake-necked turtle genus Hydromedusa"

_PeerJ, doi:10.7717/peerj.16712_

## Round 0.1 · original submission · Major Revisions

Dear Respected Authors,

Upon a thorough examination of the evaluations provided by three distinct and independent reviewers, I find the content of your manuscript to be both compelling and deserving of publication. While one reviewer expressed reservations leading to rejection, it is noteworthy that the remaining two reviewers have identified potential for improvement through their insightful feedback—ranging from major adjustments to minor refinements.

To advance toward the publication goal, it is imperative to address the revisions prescribed by the reviewers. By meticulously implementing these recommendations, we can enhance the manuscript's quality and scholarly merit. Moreover, it is evident that a consensus has emerged among all reviewers, advocating for a comprehensive revision of the introduction, materials and methods, and discussion section.

Your dedication to refining your work in accordance with the reviewers' guidance underscores your commitment to producing a publication of excellence.

Kind regards,

Armando Sunny

Reviewer 1 ·

Basic reporting

The study is well structured and provides new data to address the study of the distribution and overlap of species, in this case focused on two turtles. The manuscript is clearly written in professional, unambiguous language.
There is a little weakness, in the drafting of the aim of the study. This one is focused in the premise that the studied species share the same ecological niche; formally this is not the case (only an overlap is supported in the introduction). I suggest changing the wording.
The same goes for the hypothesis. There are no bases that allow proposing a hypothesis that suggests that both species have the same ecological niche. La specialization differentiated in the habitat use and ecological niche for both species are clear in in the presented literature.
The title of the article does not accurately describe the objective set for the study. However, with the information presented it is better to focus the objective according to the title presented.

Experimental design

The authors outline a robust and updated methodological process to compare the ecological niche of the hydromedusa species with statistical support. However, in order to highlight the contribution of the writing, I recommend including in the results the ranges of the variables used (climatic or slope) that they influence the selection of the niche in areas where they do not overlap and overlapping areas. As well as mentioning the overlapping regions, according to their results.
To develop the ecological niche models, the authors used climatic variables that represent the period from 1970 to 2000. However, the presence data of the species exceeds this range (data until 2021), therefore, the records temporarily do not coincide with variables. It is important to use variables that correspond in time. These variables are already available in the climatic platforms.
In ecological niche modeling, it is important to take care that the variables used come from similar spatial scales, in order to make them comparable. In this case, they use weather data with 5 km pixels and altitude and slope data of 90 m. I suggest using similar scales, also available on the platforms used.
I recommend that the authors highlight that biases can exist when generating pseudoabsence data, especially for models such as GLM, GAM and MARS. Also, it should be noted that formally the ensembles of all these models is not the most appropriate, since, as we colloquially say, "pears are being combined with apples" (that is, regression models, distances, artificial intelligence, etc.).

Validity of the findings

In order to improve the validity of the models, the authors can report if an external evaluation was carried out with independent data. If you don't have it, implement it, if possible.
Mention the minimum values of AUC and TSS considered to be able to include the models in the set. In addition, the performance of the different model.

Additional comments

I suggest describing the acronyms and abbreviations before using it: on line 164, PCA-env
Line 213 and 221: important
IMPORTANT: In figure 2 include in legend de code colors for each specie.
IMPORTANT: In figure 3 ¿the maps are the ensemble results? Mention it in the legend.
To resalt the study focus can be usefulness to include an overlapping map.
In the supplementary material, I have one doubt: ¿Just one record was obtained in fieldwork? If it is true, it is not relevant to mention the field work.

Reviewer 2 ·

Basic reporting

In this manuscript, the authors have proposed to identify niche similarity between two sister species, Hydromedusa maximiliani and H. tectifera. Although studies focused on testing ecological similarities between species may improve our knowledge of which abiotic and/or biotic environmental factors drive the biodiversity of geographic patterns, the manuscript suffers from several important flaws. Firstly, there is not an extensive revision of theoretical framework of ecological niche concepts, consequently, the Authors proposed an odd objective (Lines 88-90) “the objective of this study is to determine if Hydromedusa maximiliani and H. tectifera share the same ecological niche” In addition, it is observed that the authors have no experience in the use of methodological and analytical framework of ecological niche modelling approaches, as well as a lack of knowledge of their theoretical basis for their interpretation. Therefore, the authors’ findings were doubtful.

In the introduction is observer a week context, as a result of a limited literature review.
In lines 48-53 Authors mentioned “The distribution patterns of species can be shaped by various abiotic factors such as climate and elevation (Garzón, Dios & Ollero, 2008; McVicar & Körner, 2013), as well as biotic factors such as the presence of sister species in the same area (Nogueira et al.,51 2019). Certainly, the combination of several biotic and abiotic factors determines the ecological niche of the species. However, the authors present a reductionist view of the role of biotic factors and point out only the presence of sister species as the main biotic factor-driven species distribution. There is a vast literature on the theoretical basis of species ecological niches, on the role of biotic and biotic factors on species distribution, and how these factors may influence species geographical coexistence. Therefore, the authors must improve their theoretical framework. Here, there e is a list of potential references to review
• Hutchinson, G. E. (1978). An introduction to population ecology
• Soberón, J. (2007). Grinnellian and Eltonian niches and geographic distributions of species.
• Soberón, J., & Nakamura, M. (2009). Niches and distributional areas: Concepts, methods, and assumptions.
• Wisz, M. S., Pottier, J., Kissling, W. D., Pellissier, L., Lenoir, J., Damgaard, C. F., ... & Svenning, J. C. (2013). The role of biotic interactions in shaping distributions and realised assemblages of species: implications for species distribution modelling. Biological reviews, 88(1), 15-30.
• Soberón, J., & Townsend Peterson, A. (2005). Interpretation of models of fundamental ecological niches and species’ distributional areas. Biodiversity Informatics
• González-Salazar, C., Stephens, C. R., & Marquet, P. A. (2013). Comparing the relative contributions of biotic and abiotic factors as mediators of species’ distributions. Ecological Modelling, 248, 57-70.
• Lewis, J. S., Farnsworth, M. L., Burdett, C. L., Theobald, D. M., Gray, M., & Miller, R. S. (2017). Biotic and abiotic factors predicting the global distribution and population density of an invasive large mammal. Scientific reports, 7(1), 44152.
• Araújo, M. B., & Guisan, A. (2006). Five (or so) challenges for species distribution modelling. Journal of biogeography, 33(10), 1677-1688.

In lines 64-66 I have found odd Authors claimed that: “Despite the potential niche overlap, sister species may exhibit differences in the specific ecological characteristics they use to occupy similar niches (Duré, Kehr & Schaefer, 2009)”. Sister species should always exhibit minor/mayor niche overlap (niche conservatism hypothesis); therefore, they occupy similar niches in some grades, but rarely identical niches; thus, sister species exhibit ecological differences.
(Wiens et al., 2010. Ecology Letters). Thus, simply testing for sister species niche similarity itself is not of particular interest, as the result is obvious. The authors should conduct a better review of the topic to propose a novel research question in which niche similarity tests need to be implemented.
In addition, from lines 71 to 87, the authors provide a wide description of the environments where Hydromedusa species occur and where they coexist. Therefore, the results of their similarity tests are obvious (lines 226-231) and do not contribute anything new to what they are already describing about their species in the manuscript introduction

Experimental design

As mentioned above, the authors’ research question was not relevant or meaningful. This does not contribute novel knowledge. Consequently, their results, discussion, and conclusions do not contribute to filling any information gaps regarding these turtle species

Validity of the findings

As the author’s manuscript presents a weak theoretical framework and an irrelevant research question, the discussion is limited to a description of the results. In addition, the authors described how environmental variables influence Hydromedusa species distribution based on scientific literature (lines 265-292), but this is disconnected from the authors’ main goal and seems like just part of the introduction.
However, the conclusions are not well stated. For instance: Lines 332-33 “Our study reveals that Hydromedusa maximiliani and H. tectifera generally exhibit an allopatric distribution pattern, with a similar but not equivalent niches”. However, species allopatric distribution is well documented, even as described in this manuscript; therefore, this is not a novel finding of this work. In addition, niche similarity was an expected result. Lines 332-333 “However, their distinct spatial occurrence may also be influenced by habitat preference and morphology”. The same, this is documented in the literature and is not derived from the results of this work Lines 333-334 “Environmental variables, such as temperature, precipitation, and slope, play a crucial role in shaping their niche and distribution”. It is obviously, these were the variables used in ecological modeling process by Authors. So, is not a conclusion. In summary, there is not valid conclusions is this work.

Reviewer 3 ·

Basic reporting

I think this work could go beyond showing a distribution pattern in response to the species' environmental preferences. For example, how could these similarity/equivalence tests contribute to exploring the mechanism of speciation? See, for example:
https://doi.org/10.1111/j.0014-3820.2006.tb01893.x
https://doi.org/10.1007/s11692-023-09603-6
https://doi.org/10.1007/978-3-030-31167-4_26
Ideally, the introduction should show what is known about speciation patterns and the possible effect of environmental barriers in the region. These antecedents could be highly relevant to generate predictions that could be tested through niche modeling and the generation of null models. Unfortunately, the authors have provided so little information on other taxonomic groups and the biogeographic history of the region that it is difficult to understand the context of the study.
The authors should clarify the concept of sympatry that they are using and how they distinguish it from parapatry and allopatry. This is important to me since the observed and potential distribution moves between parapatry and sympatry, but it is unclear why they mention an allopatric pattern.
See, for example: https://doi.org/10.1146/annurev.es.22.110191.000315

Experimental design

The methods could be better documented. Some aspects are particularly relevant to me when generating the potential distribution maps and using the equivalence and similarity tests. The authors must mention the area used to calibrate their ecological niche models. Did you define an area for each species or the same area for both? Why did they use only 500 pseudo-absence points in such a large and heterogeneous region? The authors mention that they converted the final maps to binary but need to say what threshold rule they used, and Figure 3 does not show the binary maps but continuous ones. Please, specify why. When testing equivalence or similarity, the authors must pinpoint the exact background data they used.

Validity of the findings

The authors mention in their results the following:
“... when analyzing the niche similarity between species, the niche was found to be similar when comparing the niche of H. maximiliani to the background of H. tectifera (p = 0.01), but not similar when comparing the background of H. maximiliani is to H. tectifera (p = 0.35) (Fig. 4).“
These types of results using the similarity test are worthy of further discussion because they could even cause a lot of confusion. Knowing this, the authors who proposed these tests have designed two alternatives for the similarity tests (symmetric and asymmetric options). I suggest the authors further discuss this result and the possible methodological or biological reasons.
The authors mention in the abstract and even the conclusions that morphology (and other aspects not examined) could play an essential role in the pattern of differences/similarities. That is far from part of this study and should not be part of a conclusion.

---

## Round 0.2 · Minor Revisions

Dear Authors,

Following a meticulous review process, we greatly appreciate your dedicated efforts in addressing the valuable comments provided by the reviewers. It has been noted that some minor revisions are still necessary, particularly in the realm of materials and methods. Additionally, it appears that there might have been an oversight during the figure loading process, as the corrections based on one reviewer's feedback have not been implemented yet.

We extend our heartfelt gratitude for your continued commitment to refining your work. Your attention to these details is instrumental in ensuring the quality and accuracy of the manuscript.

Warm regards,

Armando Sunny

Reviewer 1 ·

Basic reporting

In general, the observations were attended.

Experimental design

In general, the observations were attended.

Validity of the findings

In general, the observations were attended. I request to include the corrected figures

Additional comments

The authors mention that the figures were corrected, however, they do not attach the correct figures in the new version. It is requested to integrate the corrections into the figures.

Reviewer 3 ·

Basic reporting

The authors have considered the vast majority of comments and suggestions, which is appreciated and recognized since this version is much better structured. The authors have considered the vast majority of comments and suggestions, which is appreciated and recognized since this version is much better structured. This version presents a more extensive literature review grounded in the study's context. It could be improved by placing more emphasis on the fact that this study is based on coarse-scale variables (i.e., Grinellian niche) since it seems clear that in the case of both species of turtles, fine-scale factors (e.g., competition) could play a more critical role (e.g., food resources, microhabitat, etc.).

Experimental design

This revised version has improved the documentation of the analyses and their design. I suggest clarifying the reason for both hypothesis tests (i.e., equivalence and similarity). Both have their reason for being. However, the inertia is doing both without really justifying each of them. See in detail Warren et al. (2008) for a better understanding of the two tests. Furthermore, it is essential to highlight the red flags of this type of analysis, such as the selection of environmental variables and the extent of the study area, since they tend to determine the results.

Validity of the findings

Please, see my comment above (Experimental design)

---

## Round 0.3 · accepted · Accept

Dear authors,

It brings me great pleasure to inform you that your captivating manuscript has been accepted for publication in PeerJ.

Best regards,

Armando Sunny

Reviewer 3 ·

Basic reporting

The comments were taken into account.

Experimental design

The comments were taken into account.

Validity of the findings

The comments were taken into account.

Additional comments

NA